# Radical-mediated C-C cleavage of unstrained cycloketones and DFT study for unusual regioselectivity

Mingyang Wang[1,3], Man Li[2,3], Shan Yang[1,3], Xiao-Song Xue [2]*, Xinxin Wu[1] & Chen Zhu [1]*

The C-C σ-bond activation of unstrained cycloketones represents an ingenious and advanced technique in synthetic chemistry, but it remains a challenging area which has been largely underexplored. Herein we report an efficient strategy for the direct C-C cleavage of cyclohexanones and cyclopentanones. The cyclic C-C σ-bond is readily cleaved under mild conditions with the aid of an in situ formed side-chain aryl radical. Density functional theory calculations are carried out to shed light on the unusual regioselectivity of C-C bond cleavage. The reaction affords a variety of structurally diverse 3-coumaranones and indanones that widely exist in natural products and bioactive molecules, illustrating the synthetic value of this method.

[1] Key Laboratory of Organic Synthesis of Jiangsu Province, College of Chemistry, Chemical Engineering and Materials Science, Soochow University, 199 Ren-Ai Road, 215123 Suzhou, Jiangsu, China. [2] State Key Laboratory of Elemento-organic Chemistry, College of Chemistry, Nankai University, 300071 Tianjin, China. [3]These authors contributed equally: Mingyang Wang, Man Li, Shan Yang. *email: xuexs@nankai.edu.cn; chzhu@suda.edu.cn

The C–C σ-bond is ubiquitous that constitutes the framework of organic compounds; therefore, the direct elaboration of C–C σ-bond into other valuable chemical bonds represents an ideal, atom- and step-economic synthetic tactic[1–9]. However, C–C σ-bonds usually remain inert in the transition metal-catalyzed reactions attributed to the poor interaction between the catalytic metal center and the C–C σ-orbital induced by the steric congestion and highly oriented nature of C–C σ-bonds[10]. Taking advantage of ring-strain relief, over the past few decades, small-sized rings such as cyclopropane and cyclobutane derivatives have served as privileged precursors in the transition metal-catalyzed C–C activation[11–20]. This feature has also been applied to the alkoxy-radical-promoted β–C–C bond scission that offers an efficient pathway to cleave the cyclic C–C σ-bonds of cyclopropanols and cyclobutanols[21–27]. In contrast, the C–C activation of unstrained cyclic skeletons, particularly five- and six-membered rings, is still confronted with a formidable challenge[28–31].

As fundamental structural units, cyclohexanone and cyclopentanone are readily available in numerous natural products and fine chemicals. With the exception of classic ring-expansion reactions such as the Bayer−Villiger oxidation and Schmidt reaction, however, the C–C activation of cyclohexanone and cyclopentanone has been rarely explored due to the significantly decreased ring-strain energy and remarkable stability (Fig. 1a)[32]. Recently, Dong et al. developed an elegant strategy based on the installation of temporary directing group to activate the C–C bond of cyclopentanones through transition metal-catalysis (Fig. 1b)[33–35]. But the strategy was less efficient for C–C activation of the relatively more stable cyclohexanones. Herein, we disclose an efficient strategy for the direct C–C cleavage of unstrained cycloketones. With the assistance of the side-chain aryl radical, both cyclohexanones and cyclopentanones are readily cleaved, leading to a variety of structurally diverse 3-

coumaranones and indanones (Fig. 1c). Notably, these structural motifs are widely found in natural products and pharmaceuticals, manifesting the synthetic value of this protocol (Fig. 1d). Mechanistically, the cleavage of C–C σ-bond via path **b** to form the primary alkyl radical (Int. **b**) is unusual[36,37], as the generation of ring-expansion Int. **a** via path **a** is supposed to be more favored due to the thermodynamically stability of secondary alkyl radical and p-π conjugate effect provided by the adjacent oxygen atom according to the Dowd−Beckwith ring-expansion reaction[38–47]. Thus, density functional theory (DFT) as well as experimental studies have been sought to address the perplexing regioselectivity of C–C bond cleavage.

## Results

**Reaction parameters survey.** The project commenced with an extensive survey of reaction parameters for the C–C activation of 2-(2-iodophenoxy) cyclohexanone **1a** under radical conditions (Table 1). It was found that with PhCF$_3$ as solvent the combination of AIBN and TTMSS could abstract the iodine atom to generate aryl radical that triggered the subsequent C–C cleavage, affording 3-coumaranone **2a** in modest yield (Table 1, entry 1). An alternative pathway, the addition of aryl radical to cyclohexanone followed by dehydration, led to the benzofuran **2a′** that was identified as main byproduct in the reaction. While assessing the effect of base, it was surprising that the hydrated K$_2$HPO$_4$ delivered a good yield (Table 1, entry 2), whereas the anhydrous one as well as other bases only afforded a trace amount of products (Table 1, entries 3–4). This prompted us to turn our attention to examining the influence of water (Table 1, entries

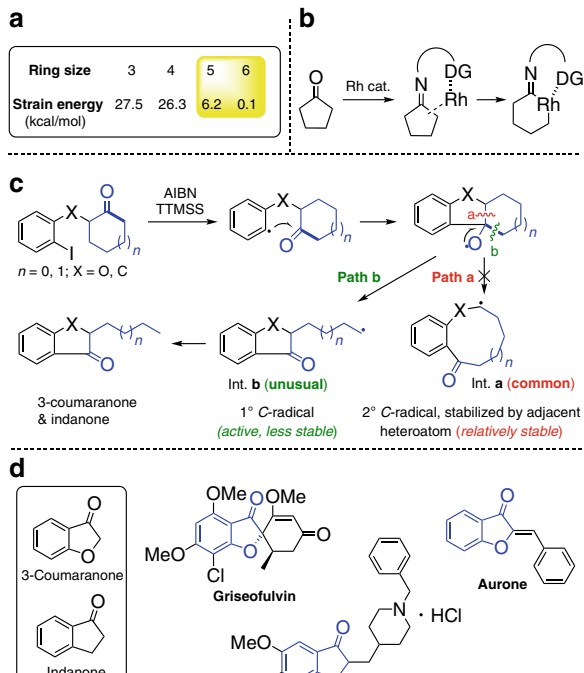

**Fig. 1 Challenge and synthetic value for C–C cleavage of cyclohexanone and cyclopentanone. a** Ring strain energy. **b** Activation of cyclopentanone by temporary directing mode by Dong et al. **c** Aryl radical-promoted activation of both cyclohexanone and cyclopentanone. **d** Natural products and drugs containing 3-coumaranone and indane.

**Table 1 Reaction parameters survey.**

| Entry[a] | Additive | Solvent | Yield of 2a (%)[b] |
|---|---|---|---|
| 1 | None | PhCF$_3$ | 66 |
| 2 | K$_2$HPO$_4$·3H$_2$O (1 equiv.) | PhCF$_3$ | 76 |
| 3 | K$_2$HPO$_4$ (1 equiv.) | PhCF$_3$ | <10 |
| 4 | K$_2$CO$_3$ or K$_3$PO$_4$ (1 equiv.) | PhCF$_3$ | <10 |
| 5 | H$_2$O (3 equiv.) | PhCF$_3$ | 77 |
| 6 | H$_2$O (20 equiv.) | PhCF$_3$ | 85 |
| 7 | H$_2$O (100 equiv.) | PhCF$_3$ | 75 |
| 8 | H$_2$O (20 equiv.) | DMF | 25 |
| 9 | H$_2$O (20 equiv.) | DMSO | 16 |
| 10 | H$_2$O (20 equiv.) | THF | 26 |
| 11 | H$_2$O (20 equiv.) | DCE | 61 |
| 12 | H$_2$O (20 equiv.) | Dioxane | 65 |
| 13[c] | H$_2$O (20 equiv.) | PhCF$_3$ | 84 |
| 14[d] | H$_2$O (20 equiv.) | PhCF$_3$ | 66 |
| 15[c,e] | H$_2$O (20 equiv.) | PhCF$_3$ | 74 |
| 16[c,f] | H$_2$O (20 equiv.) | PhCF$_3$ | 56 |
| 17[g] | None | PhCF$_3$ | 78 |
| 18[c,g] | None | PhCF$_3$ | 31 |

[a]Reaction conditions: **1a** (0.2 mmol), TTMSS (0.3 mmol, 1.5 equiv.), AIBN (0.04 mmol, 20 mol %), and additive in solvent (2 mL) at 80 °C.
[b]Yield of isolated product.
[c]AIBN (0.01 mmol, 5 mol %).
[d]Use of air instead of AIBN.
[e]70 °C.
[f]PhCF3 (4 mL).
[g]Bu3SnH instead of TTMSS.

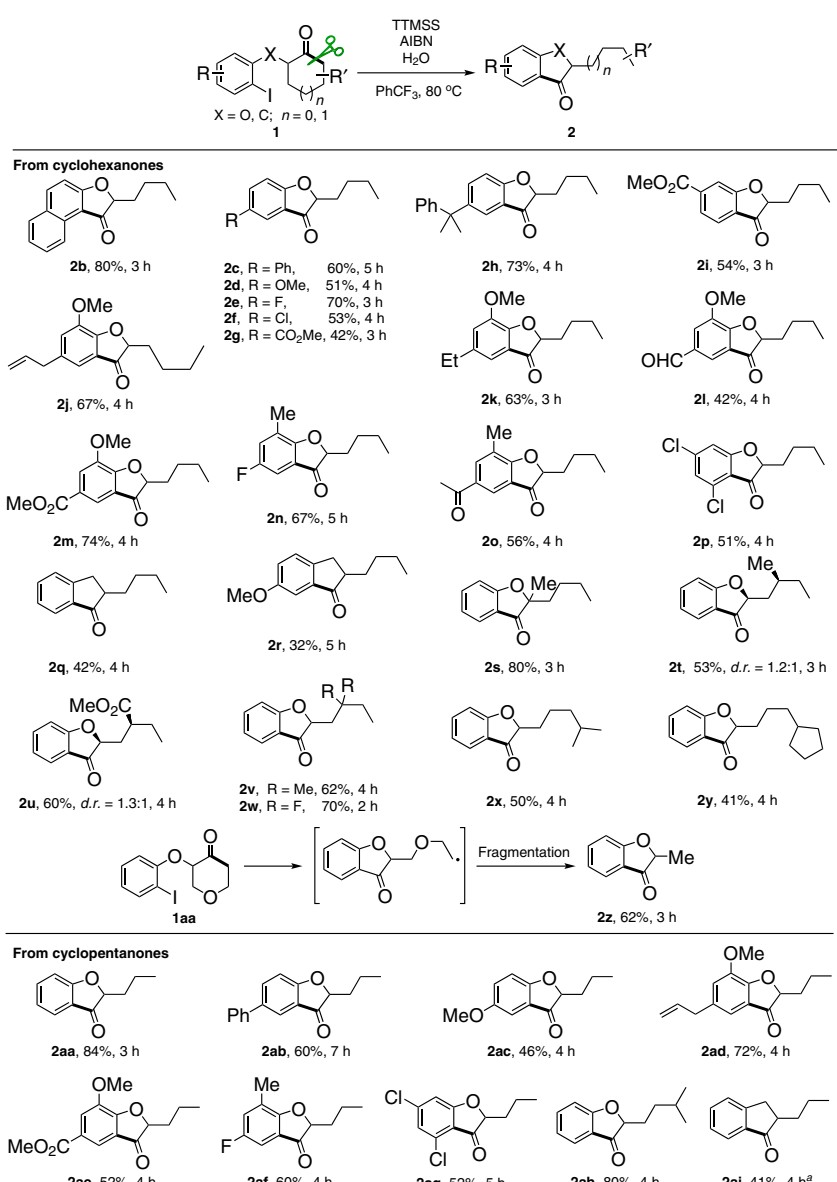

**Fig. 2 Generality of protocol and scope of 3-coumaranone and indanone.** Reaction conditions: **1** (0.2 mmol), TTMSS (0.3 mmol, 1.5 equiv.), AIBN (0.01 mmol, 5 mol %), and H$_2$O (4 mmol, 20 equiv.) in PhCF$_3$ (2 mL) at 80 °C. Yields of isolated products are given. [a]**1ai** (0.3 mmol), TTMSS (1.5 equiv.), V-40 (20 mol %), and K$_2$HPO$_4$·3H$_2$O (1 equiv.) in PhCF$_3$ (6 mL) at 110 °C.

5–7). Indeed, the addition of 20 equiv. of water increased the yield to 85% (Table 1, entry 6). Solvent screening revealed that PhCF$_3$ was more efficient than other solvents (Table 1, entries 8–12). Reducing the loading of AIBN to 5 mol % resulted in a similarly high yield (Table 1, entry 13). Notably, the reaction could be carried out under air without the use of AIBN, leading to a synthetically useful yield (Table 1, entry 14). Decrease of reaction temperature compromised the reaction outcome (Table 1, entry 15). Reducing the concentration also decreased the reaction yield (Table 1, entry 16). Although using Bu$_3$SnH instead of TTMSS also led to good yield (Table 1, entry 17, with 20 mol % of AIBN; Table 1, entry 18, with 5 mol % of AIBN), the use of less toxic silane is still preferred in the reaction.

**Scope of substrates**. With the optimized reaction conditions in hand, we set about evaluating the generality of the protocol (Fig. 2). A variety of 2-substituted cyclohexanones (**1b**–**1z**) were firstly examined. All the reactions were readily accomplished through the regiospecific cyclic C–C σ-bond scission and

subsequent C–C bond formation within a few hours. Both electron-rich and -deficient aryl substituents were well tolerated in the reaction, affording the corresponding 3-coumaranones **2**. Under the identical conditions, the naphthyl-fused coumaranone **2b** was also obtained in a high yield. The protocol could furnish a diversity of multiple functionalized 3-coumaranones (**2j**–**2p**) even bearing susceptible groups such as allyl (**2j**) and formyl (**2l**). Indanones (**2q**–**2r**) could be prepared by this method, albeit in relatively lower yields. Notably, the use of 2,2-disubstituted cyclohexanone **1s** did not reverse the regioselectivity, though the competitive ring-expansion pathway seemed more likely due to the formation of thermodynamically favored tertiary alkyl radical. Substitutions on the cyclic framework did not influence the transformation, leading to the corresponding 3-coumaranones (**2t**–**2y**) with various aliphatic side chains. Interestingly, the reaction with tetrahydropyranone **1z** resulted in 2-methyl-3-coumaranone **2z** probably after the fragmentation of the ring-opened radical intermediate (see Supplementary Fig. 12). Some representative examples of cyclopentanones (**1aa**–**1ai**) were then

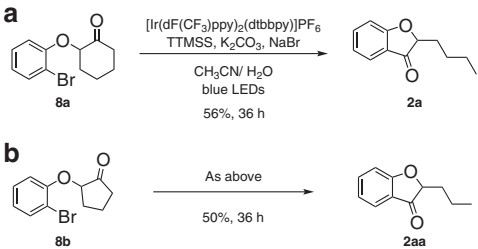

**Fig. 3 Transformation of arylbromide analogs. a** Reaction with 2-(2-bromophenoxy) cyclohexanone. **b** Reaction with 2-(2-bromophenoxy) cyclopentanone.

**Fig. 4 Modification of complex natural product. a** Reaction with Estrone derivative **3a**. **b** Reaction with Estrone derivative **3b**.

investigated. The substitution effect on either arene or cyclopentanone was examined. Likewise, the reaction readily proceeded regardless of the electronic and steric characters, affording the resultant 3-coumaranones (**2aa**–**2ah**) in synthetically useful yields. The corresponding indanone (**2ai**) could also be obtained by this method, but with the modified conditions. Notably, ring opening of cycloheptanone could also afford the corresponding product **2aj** in 30% yield (see Supplementary Fig. 13).

The conversion of 2-(2-bromophenoxy) cyclohexanone **8a** and cyclopentanone **8b** could not occur under the current condition due to the lower reactivity of arylbromide. However, the expected transformation proceeded with a modified photochemical condition, affording the corresponding products in synthetically useful yields (Fig. 3).

The practicality of the method was further illustrated in the modification of complex natural products (Fig. 4). Despite the location of iodine atom, both the estrone derivative **3a** and **3b** were readily converted to the corresponding coumaranones **4a** and **4b** in synthetically useful yields, respectively.

**Mechanistic studies**. To shed light on the unusual regioselectivity of the cyclic C–C cleavage, we conducted computational studies using DFT calculations (for details, see the SI). The results are presented in Fig. 5. The cleavage of the C1–C2 bond in **Int.1** via **TS1a** (Fig. 5a, path a) to give the nine-membered ring radical intermediate (**Int.a**) is a facile and reversible process with a barrier of only 3.2 kcal/mol. The energy barrier is 3.4 kcal/mol higher for the cleavage of the C1–C3 bond (via **TS1b**, path b) to generate the primary alkyl radical intermediate **Int.b**. As shown in Fig. 5b, the natural bond orbital (NBO) analysis of **Int.1** reveals the hyperconjugation between the lone pairs at O4 ($n_{O4}$) and the antibonding orbitals of the C1–C3 and C1–C2 bonds ($\sigma^*$ C1–C3 and $\sigma^*$ C1–C2), which would weaken both C–C single bonds and thus facilitate their cleavage[48]. The $n_{O4} \rightarrow \sigma^*$ C1–C2 interaction

was calculated to be about 7 kcal/mol stronger than the $n_{O4} \rightarrow \sigma^*$ C1–C3 interaction (see Supplementary Tables 1 and 2). This may explain why the cleavage of the C1–C2 bond exhibits a lower energy barrier.

Interestingly, **Int.a** is 1.8 kcal/mol less stable than **Int.b**, presumably due to a greater ring strain in the nine-membered ring. The subsequent hydrogen-atom abstraction is irreversible and requires a higher barrier than the cleavage of C–C bond. The barriers for hydrogen-atom abstraction from TTMSS by **Int.a** and **Int.b** are 15.2 (via **TS2a**) and 10.9 (via **TS2b**) kcal/mol, respectively, which are at least 4.3 kcal/mol higher than that for the cyclic C–C cleavage. The formation of 3-coumaranone **2a** is thermodynamically favored over 3,4,5,6-tetrahydrobenzo[b]oxonin-7(2 H)-one **2a′** by 12.8 kcal/mol. Accordingly, the 3-coumaranone **2a** is both kinetically and thermodynamically favored product, which is consistent with experimental observation that only **2a** was observed. Notably, an intramolecular hydrogen-atom abstraction (1,5-H transfer) is also possible for **Int.b**. The calculations show that the intramolecular 1,5-hydrogen atom abstraction proceeds via **TS3b** has a barrier of 10.4 kcal/mol, leading to a tertiary carbon radical intermediate **Int.2**, which is quite stable and located 26.2 kcal/mol below **Int.1**. **Int.2** abstracts a hydrogen atom from TTMSS to deliver the final product via **TS4b** with a barrier of 19.8 kcal/mol. Since the intramolecular 1,5-hydrogen atom abstraction is slightly favored (only 0.5 kcal/mol) over the intermolecular hydrogen-atom abstraction (**TS2b** vs. **TS3b**), the two processes could compete with each other. This corresponds well with the isotope-labeling experiments (Fig. 6). It could also be speculated that **Int.1**, **Int.a**, and **Int.b** are in an equilibrium and a high energy barrier for H-transfer from TTMSS for cyclic radical **Int.a** may allow for preferential quenching of primary alkyl radical **Int.b** via either intramolecular 1,5-H-shift or intermolecular H-shift from TTMSS. It may be possible for alkoxy radical **Int.1** to abstract hydrogen from TTMSS to form alcohol. Dehydration of the product can give the benzofuran-type byproduct **2a′**. Overall, the calculations revealed that the selectivity of the reaction is mainly controlled by the step of hydrogen-atom abstraction instead of the C–C bond cleavage. The primary alkyl radical is more reactive than the secondary alkyl radical for hydrogen atom abstraction, leading to the observed selectivity. Replacing TTMSS by tin hydride as hydrogen atom donor leads to the similar calculational outcome (see Supplementary Fig. 178). The rational analysis for the formation of indanones could also be obtained from the DFT studies (see Supplementary Fig. 179).

To provide support for the computational results, we further carried out the experiments with the deuterium-labeled materials. Firstly, the reaction of **1a** in the presence of *d*-TTMSS delivered product **5** with H/D ratio as 0.7:0.3, indicating that the final step of H-abstraction could proceed via either the intramolecular 1,5-HAT or intermolecular H-abstraction from TTMSS (Fig. 6a). While employing the deuterated cyclohexanone **6** under the previous conditions, the counting of H and D in product **7** also suggested that the radical chain termination step could go through both two pathways (Fig. 6b).

A further attempt was conducted to elucidate the effect of water (for details, see Supplementary Figs. 8–10). After the completion of reaction, a highly acidic water drop (pH < 1, presumably the aqueous solution of HI) precipitated in the reaction vial, which was obtained by hydrolysis of the byproduct (TMS)₃SiI[49,50]. Moreover, the addition of TFA instead of water to the reaction largely suppressed the reaction outcome by increasing the formation of benzofuran byproduct **2a′**. Thus, it was hypothesized that the HI-induced acidic condition was fatal to the reaction, and water could separate HI from the reaction through the formation of a biphasic solution.

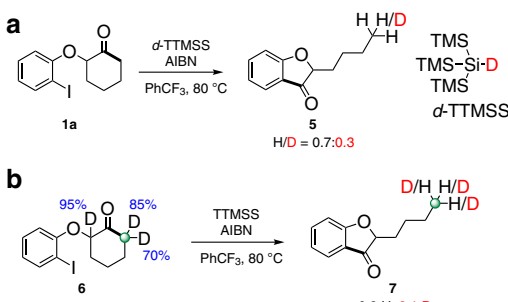

**Fig. 5 Computational study. a** Free energy diagram (kcal/mol) calculated at the (SMD)-M06-2×/6-311++G(2d,p)//M06-2×/6-31G(d) level of theory (pink: bond length in Å). $\Delta G$ calculated at 298.15 K. **b** The natural bond orbital (NBO) analysis of **Int.1**. Interaction energies are in kcal/mol.

## Methods

**General procedure for ring opening of cycloketones**. Cycloketones (**1**) (0.2 mmol, 1 equiv.) and AIBN (0.01 mmol, 5 mol %) were loaded in a reaction vial that was subjected to evacuation/flushing with $N_2$ three times. Then TTMSS (0.3 mmol, 1.5 equiv.), $H_2O$ (4 mmol, 20 equiv.) and PhCF$_3$ (2 mL) were added to the mixture which was then heated to 80 °C. After the reaction completion, the reaction mixture was concentrated in vacuo. Purification by flash column chromatography on silica gel afforded the desired product.

## Data availability

The authors declare that all other data supporting the findings of this study are available within the article and Supplementary Information files, and also are available from the corresponding author on reasonable request.

**Fig. 6 Deuterium labeling experiments. a** Reaction with d-TTMSS. **b** Reaction with deuterated cyclohexanone **6**.

## Discussion

In summary, we have disclosed an efficient radical-mediated protocol for the cleavage of inert unstrained C–C σ-bonds. Both cyclohexanones and cyclopentanones prove to be suitable substrates. A variety of synthetically valuable 3-coumaranones or indanones are readily prepared under mild conditions. The cyclic C–C bonds are cleaved in an unusual regioselectivity. DFT studies as well as the deuterium labeling experiments are sought to address the selectivity issue. This protocol provides a practical method and informative mechanistic insight for the elusive C–C activation of unstrained cycloketones.

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

## Acknowledgements

We are grateful for the financial support from the National Natural Science Foundation of China (21722205, 21971173, 21772098, and 21933004), the Project of Scientific and Technologic Infrastructure of Suzhou (SZS201708), and the Priority Academic Program Development of Jiangsu Higher Education Institutions (PAPD).

## Author contributions

C.Z. conceived and designed the experiments; M.W. and S.Y. carried out most of the experiments; M.L. and X.-S.X carried out DFT computation; M.W., M.L., S.Y., X.-S.X., X.W., and C.Z. analyzed data; X.-S.X. and C.Z. wrote the paper; C.Z. directed the project.

## Competing interests
The authors declare no competing interests.
