## [Peer Review File · Nature Communications]

Reviewers' comments:

Reviewer #1 (Remarks to the Author):

The current work reports a new experimental strategy for the direct C-C cleavage of cyclohexanones and cyclopentanones. The authors found that the cyclic C-C σ -bond is readily cleaved under mild conditions with the aid of an in situ formed side-chain aryl radical. DFT calculations were also carried out to explain the unusual regioselectivity of the C-C bond cleavage and supports the experimental observation. However, there are a number of suggestions and comments that should be addressed before the manuscript is considered to be published:

Major:

- 1) It is commonly known that DFT calculated results including energies are very sensitive to the used exchange-correlation functionals. I would like to see some extra single-point calculations with different types of functionals such as PBE, B3LYP, TPSS, etc. from which one can judge whether the calculated results are qualitatively unchanged from one functional to another.
- 2) It is known that M06 functional gives wrong electronic density, see SCIENCE, 355, 49, 2017. The related electronic density analysis and NBO could be not too accurate. Need some checks and comments.
- 3) For such small organic systems, why not add some high-level couple-cluster single point energies, just reporting DFT energies. It is a little confusing for me. To be honestly, for the time being, CCSD(T) calculations on such size of systems become feasible with small computer clusters.
- 4) If I understand correctly, the solvent effects are merely considered at the level of single point energies. I suggested adding some extra data to demonstrate that solvents effects cannot affect the optimized structures.

Minor:

1. In the line 10, there should be 'Density functional theory calculations ...' instead of 'Density functional calculations ...'.
2. In the Fig. 1A, for the words 'Ring Size' and 'Strain energy', their case does not matter, but be consistent to avoid confusion.
3. In the line 92, between the '3-coumaranones' there is an extra space.
4. In the Fig. 4, there may be less a colon behind the '(A)'.
5. In the Table S1, the 'Interaction Energies' should correct to 'Interaction energies'.
6. In the Fig. S7, does the compound 2a' is a new product? I think it may be 2a'.

Reviewer #2 (Remarks to the Author):

In this manuscript, the authors have described a radical-mediated transformations on cyclopentanone and cyclohexanone substrates to form 3-coumaranone type products. This reaction was initiated via classical radical chain chemistry (AIBN and TTMSS), supposedly proceeded through a series of radical-mediated events, an intramolecular addition which triggered the c-c bond scission to cleave the cyclohexanone moiety and eventually termination.

The C-C bond activation has drawn a lot of attention in the synthetic community lately, this approach could provide valuable opportunities to synthesize molecules. With the advancement of transition metal catalysis, substantial advancement has been achieved in this area, and elegant applications in complex molecule synthesis have been successfully demonstrated. Although the authors have acknowledged the significance in the introduction, it does not apply to the reaction described here. In classical radical chain chemistry, c-c bond scission is omnipresent, in Barton reaction etc. While reading the manuscript, one can't resist the strange, yet strong, feeling of déjà vu. I am sure the authors have done extensive literature research and no one has published the exact same thing before. Yet, every radical-mediated step of this reaction has certainly been well-

studied decades ago. The substrates are literally too specialized, a cyclic ketone with an aromatic ring (limited to 2-iodo substituted ones) tethered next to the carbonyl. The use of excess equivalent of silane, and AIBN, does not seem attractive to synthetic chemists any more. It certainly would be more interesting if the authors could use catalytic strategies to initiate the reaction.

In general, the method described herein lacks generality and practicality in application, novelty in concept. This contribution falls short to fulfill, in my opinion, the requirements expected from a contribution in Nature Comm.

Reviewer #3 (Remarks to the Author):

Zhu et al reported an efficient and novel approach for radical C-C bond cleavage of unstrained cycloketones. This method has brought a new opening in the arena of C-C activation of unstrained cyclic skeletons through radical mechanism. The scope of cycloketones used is very broad, including many 5- and 6-membered cycloketones. This is a potentially synthetically useful reaction. Furthermore, the unusual reactivity and regioselectivity in this reaction were well investigated by DFT calculations and the results were consistent with the experimental observation. The manuscript is well written with experimental results adequately documented including full characterization of the compounds prepared. Overall, the method is well-developed and evaluated, and this strategy offers a nice method for C-C bond cleavage of unstrained compounds. Given the novelty and potential utility of this transformation, the manuscript warrants publication in Nature Commun.

Some corrections should be accomplished before publication:

1. as shown in table 1, entries 3 and 4. anhydrous bases only afforded a trace amount of 2a. What the reason for these results? the authors should give some comments on it. In the main article the authors hypothesized that the HI-induced acidic condition was fatal to the reaction. It seems that base should be helpful for this transformation.
2. Some recent reviews on unstrained C-C bond involving organometallic intermediates should be cited:
Chem. Asian J. 2014, 9, 3360; Chem. Soc. Rev. 2018, 47, 7078.
3. It seems that 2-(2-iodophenoxy) cyclohexanone 1a was worked well under the standard conditions. How about the analogue brominated substrate?
4. in line 74, "AIBN (0.04 mmol, 20 mol %)" should be "AIBN (0.01 mmol, 5 mol %)"

Response to referee#1

1. Comment: *The current work reports a new experimental strategy for the direct C-C cleavage of cyclohexanones and cyclopentanones. The authors found that the cyclic C-C σ -bond is readily cleaved under mild conditions with the aid of an in situ formed side-chain aryl radical. DFT calculations were also carried out to explain the unusual regioselectivity of the C-C bond cleavage and supports the experimental observation. However, there are a number of suggestions and comments that should be addressed before the manuscript is considered to be published:*

RESPONSE: We thank the reviewer for his/her suggestions and comments. Following your suggestions, we have carefully revised our manuscript.

2. Comment: *It is commonly known that DFT calculated results including energies are very sensitive to the used exchange-correlation functionals. I would like to see some extra single-point calculations with different types of functionals such as PBE, B3LYP, TPSS, etc. from which one can judge whether the calculated results are qualitatively unchanged from one functional to another.*

RESPONSE: Thanks for this comment. We have conducted extra single-point calculations with different types of functionals including B3LYP-D3(BJ), B97X-D,

PBE0-D3(BJ), PBE-D3(BJ), and TPSS-D3(BJ). As shown below, the trend for all the results are very similar to each other, and the calculated results are qualitatively unchanged from one functional to another. These results have been added into revised SI (Fig S11).

(SMD)-M06-2X/6-311++G(2d,p)//M06-2X/6-31G(d)

(SMD)-B3LYP-D3(BJ)/6-311++G(2d,p)//M06-2X/6-31G(d)

(SMD)- ω B97X-D/6-311++G(2d,p)//M06-2X/6-31G(d)

(SMD)-PBE0-D3(BJ)/6-311++G(2d,p)//M06-2X/6-31G(d)

(SMD)-PBE-D3(BJ)/6-311++G(2d,p)//M06-2X/6-31G(d)

(SMD)-TPSS-D3(BJ)/6-311++G(2d,p)//M06-2X/6-31G(d)

3. Comment: It is known that M06 functional gives wrong electronic density, see *SCIENCE*, 355, 49, 2017. The related electronic density analysis and NBO could be not too accurate. Need some checks and comments.

RESPONSE: We thank the reviewer for this suggestion. In order to check the calculated results, we have added the NBO analysis by PBE0-D3(BJ) and B3LYP-D3(BJ). As

shown in the table, the interaction energies calculated by all the 3 methods are very close. These results have been added into revised SI (Table S2).

Table S2. The natural bond orbital (NBO) analysis of Int.1 at different theoretical calculation levels of theory. Interaction energies are in kcal/mol.

Interaction	Lone-pair electrons		Unpaired electron	In total
	α spin	β spin	α spin	
$nO_4 \longrightarrow \sigma^* C_1-C_2$				
UM06-2X/6-311++G(2d, p)	2.2	5.1	4.8	12.1
UPBE0-D3(BJ)/6-311++G(2d, p)	2.2	4.1	4.2	10.6
UB3LYP-D3(BJ)/6-311++G(2d, p)	2.3	4.0	3.6	9.9

$nO_4 \longrightarrow \sigma^* C_1-C_3$				
UM06-2X/6-311++G(2d, p)	5.1	-- [a]	-- [a]	5.1
UPBE0-D3(BJ)/6-311++G(2d, p)	4.5			4.5
UB3LYP-D3(BJ)/6-311++G(2d, p)	4.4			4.4
[a] Below the threshold of 0.5 kcal/mol.				

4. Comment: For such small organic systems, why not add some high-level couple-cluster single point energies, just reporting DFT energies. It is a little confusing for me. To be honestly, for the time being, CCSD(T) calculations on such size of systems become feasible with small computer clusters.

RESPONSE: Thanks for this suggestion. We have attempted to do CCSD(T) calculations. We found that some structures have 68 atoms, i.e. TS2a, TS2b, and TS3b., which are too expensive for CCSD(T) calculations. Given that the additional results obtained by other theoretical methods are qualitatively unchanged, our results should, at least qualitatively, be independent of computational methods.

5. *Comment:* If I understand correctly, the solvent effects are merely considered at the level of single point energies. I suggested adding some extra data to demonstrate that solvents effects cannot affect the optimized structures.

RESPONSE: Thanks. We have re-optimized all the structures in solvent and added these results into the figure below and revised SI (Fig S12). According to the calculated energies, the solvents effects did not have a significant effect on the optimized structures.

6. *Comment:*

Minor:

1. In the line 10, there should be 'Density functional theory calculations ...' instead of 'Density functional calculations ...'.
2. In the Fig. 1A, for the words 'Ring Size' and 'Strain energy', their case does not matter, but be consistent to avoid confusion.
3. In the line 92, between the '3-coumaranones' there is an extra space.

4. In the Fig. 4, there may be less a colon behind the '(A)'.
5. In the Table S1, the 'Interaction Energies' should correct to 'Interaction energies'.
6. In the Fig. S7, does the compound 2a'' is a new product? I think it may be 2a'.

RESPONSE: All these minor issues have been revised. See the updated SI and the highlighted parts in manuscript.

Response to referee#2

1. Comment: *In classical radical chain chemistry, c-c bond scission is omnipresent, in Barton reaction etc. While reading the manuscript, one can't resist the strange, yet strong, feeling of déjà vu. I am sure the authors have done extensive literature research and no one has published the exact same thing before. Yet, every radical-mediated step of this reaction has certainly been well-studied decades ago.*

RESPONSE: Thanks for the comments. Firstly, this transformation is novel, as it demonstrates unusual C-C bond scission compared to the well-studied Dowd-Beckwith reaction (this referee might get a feeling of déjà vu from this name reaction?).

Every synthetic chemist still does new chemistry by exploiting well-known elementary steps such as oxidative addition, reductive elimination, etc., however the novelty resides on whether something is expected (or logical) based on prior knowledge. As we clearly claimed, the results in this manuscript are unexpected, providing complementary knowledge to the classic Dowd-Beckwith reaction.

I would also like to say that most of the elementary reactions in radical chemistry were established half century ago, but this does not impede the area to attract a lot of attention in the synthetic community lately.

In my opinion, organic chemists should pay more attention to the origin for this unusual C-C bond cleavage and the mechanistic aspects.

2. Comment: *The substrates are literally too specialized, a cyclic ketone with an aromatic ring (limited to 2-iodo substituted ones) tethered next to the carbonyl.*

RESPONSE: Thanks for the comments. Everyone knows there is no universal reaction and every reaction is specific to a number of substrates. In this work, the substrates (a cyclic ketone with an aromatic ring tethered next to the carbonyl) are designed for the expected transformation. Likewise, in the classic Dowd-Beckwith reaction, all the substrates employ a cyclic ketone with an aliphatic chain tethered next to the carbonyl. But no one argues this name reaction is too specialized.

Moreover, in this revision, the substrates are no longer limited to 2-iodo substituted ones. The reaction of 2-(2-bromophenoxy) cyclohexanone proceeds with a modified photocatalytic conditions, see Fig 3.

3. Comment: *The use of excess equivalent of silane, and AIBN, does not seem attractive to synthetic chemists any more. It certainly would be more interesting if the authors could use catalytic strategies to initiate the reaction.*

RESPONSE: We only use 5 mol % of AIBN in the reaction.

Response to referee#3

1. Comment: *as shown in table 1, entries 3 and 4. anhydrous bases only afforded a trace amount of 2a. What the reason for these results? the authors should give some comments on it. In the main article the authors hypothesized that the HI-induced acidic condition was fatal to the reaction. It seems that base should be helpful for this transformation.*

RESPONSE: Thanks. We found acidic conditions promoted the dehydration of an intermediate to form the byproduct 2a' (Table 1) and thus decreased the yields. We agree with the reviewer that, in theory, the addition of base could remove acid from the reaction and be helpful for the reaction. However, it was found that the addition of base could also damage other reactive species other than acid, and totally suppressed the reaction.

2. Comment: Some recent reviews on unstrained C-C bond involving organometallic intermediates should be cited: *Chem. Asian J.* 2014, 9, 3360; *Chem. Soc. Rev.* 2018, 47, 7078.

RESPONSE: Done, see ref. 8 and 9.

3. Comment: It seems that 2-(2-iodophenoxy) cyclohexanone **1a** was worked well under the standard conditions. How about the analogue brominated substrate?

RESPONSE: Thanks for the suggestion. The reaction of 2-(2-bromophenoxy) cyclohexanone was not good under the standard conditions. However, after a systematic survey (Tables below), it now works with a modified photocatalytic conditions to give the expected product in useful yields. Also see Fig 3.

Entry	H-source (equiv)	Base (equiv)	T (°C)	H ₂ O	Yield
1	TTMSS (1.5)	K ₂ CO ₃ (1.5)	80	0.2 mL	14.5%
2	TTMSS (1.5)	K ₂ HPO ₄ (1.5)	80	0.2 mL	16.6%
3	TTMSS (1.5)	K ₃ PO ₄ (1.5)	80	0.2 mL	19%
4	TTMSS (1.5)	K ₂ HPO ₄ (1.5)	90	0.2 mL	<10%
5	TTMSS (1.5)	K ₂ HPO ₄ (1.5)	120	0.2 mL	<10%
6	n-Bu ₃ SnH (1.2)	-	80	0.2 mL	<10%
7	n-Bu ₃ SnH (1.2)	K ₂ HPO ₄ (1.5)	80	0.2 mL	<10%
8	n-Bu ₃ SnH (1.2)	K ₂ HPO ₄ (1.5)	120	20 mL	<10%
9	Ph ₃ SiH (1.5)	K ₂ HPO ₄ (1.5)	80	20 mL	<10%
10	PhMe ₂ SiH (1.5)	K ₂ HPO ₄ (1.5)	80	20 mL	<10%
11	Ph ₂ MeSiH (1.5)	K ₂ HPO ₄ (1.5)	80	20 mL	<10%
12	i-Pr ₃ SiH (1.5)	K ₂ HPO ₄ (1.5)	80	20 mL	<10%

13 Et₃SiH (1.2) K₂HPO₄ (1.5) 80 20 mL <10%

P.C (A): [Ir(dF(CF₃)ppy)₂(dtbbpy)]PF₆

P.C (B): [Ir(ppy)₂(dtbbpy)]PF₆

Entry	P.C. (equiv)	H-source (equiv)	Base (equiv)	Additive (equiv)	solvent	hv	Yield
1	A (3%)	TTMSS (1.5)	K ₂ CO ₃ (2.0)	NaBr (2.0)	CH ₃ CN (4 mL)	15 W	47%
2	A (3%)	TTMSS (1.5)		NaBr (2.0)	CH ₃ CN (4 mL)	15 W	50%
3	A (3%)	TTMSS (1.5)	K ₂ CO ₃ (2.0)	-	CH ₃ CN (4 mL)	15 W	30%
4	Ir(PPy) ₃ (3%)	TTMSS (1.5)	K ₂ CO ₃ (2.0)	NaBr (2.0)	CH ₃ CN (4 mL)	15 W	40%
5	B (3%)	TTMSS (1.5)	K ₂ CO ₃ (2.0)	NaBr (2.0)	CH ₃ CN (4 mL)	15 W	40%
6	Ru(bPy) ₃ Cl ₂ ·6H ₂ O (3%)	TTMSS (1.5)	K ₂ CO ₃ (2.0)	NaBr (2.0)	CH ₃ CN (4 mL)	30 W	<10%
7	Ir(PPy) ₃ (3%)	TTMSS (2.0)	K ₂ CO ₃ (2.0)	NaBr (2.0)	CH ₃ CN (4 mL)	30 W	55%
8	A (3%)	TTMSS (2.0)	K ₂ CO ₃ (2.0)	NaBr (2.0)	CH ₃ CN (2 mL)	30 W	28%
9	A (3%)	TTMSS (2.0)	K ₂ CO ₃ (2.0)	NaBr (2.0)	CH ₃ CN (8 mL)	30 W	34%
10^a	A (3%)	TTMSS (2.0)	K₂CO₃ (2.0)	NaBr (2.0)	CH₃CN (4 mL)	30 W	56%
11 ^a	Ir(PPy) ₃ (3%)	TTMSS (2.0)	K ₂ CO ₃ (2.0)	NaBr (2.0)	CH ₃ CN (4mL)	30 W	trace

^aH₂O (0.2 mL)

P.C (A): [Ir(dF(CF₃)ppy)₂(dtbbpy)]PF₆

Entry	P.C. (equiv)	H-source (equiv)	Base (equiv)	Additive (equiv)	solvent	hv	Yield
1^a	A (3%)	TTMSS (2.0)	K₂CO₃ (2.0)	NaBr (2.0)	CH₃CN (4 mL)	30 W	50%
2	Ir(PPy) ₃ (3%)	TTMSS (2.0)	K ₂ CO ₃ (2.0)	NaBr (2.0)	CH ₃ CN (4 mL)	30 W	40%

^aH₂O (0.2 mL)

4. Comment: in line 74, “AIBN (0.04 mmol, 20 mol %)” should be “AIBN (0.01 mmol, 5 mol %)”

RESPONSE: No mistake in this issue. We initially screened the conditions with 20 mol % of AIBN, then it was found the amount of AIBN could be reduced to 5 mol %.

Response to Editor

Comment: the scope might be considerably improved based on the reviewer’s comments, for example, with bromoaryl substituents instead of iodoaryl, medium-sized unstrained cycloketones (7- or 8-membered rings), with further use of the alkyl handle resulting from the C-C scission, etc.

RESPONSE: The bromoaryl substituents (containing bromoaryl cyclopentanone and cyclohexanone) could not work under the previous conditions, but now they work in a modified condition to give the products in useful yields. Medium-sized cycloketones, especially 7- and 8-membered rings, are always challenging substrates due to the transannular strain energy. In fact, they are scarcely employed in ring-opening reactions in previous reports. For 7-membered cycloketone, now we could obtain 30% yield of product with a modified condition (see Page 6, and also Fig S7); but opening of 8-membered cycloketone still failed.

REVIEWERS' COMMENTS:

Reviewer #3 (Remarks to the Author):

In this revised version, Xue, Zhu and co-workers have carefully addressed all concerns of the previous reviewers and have adopted their suggestions. Some extra single-point calculations with different types of functionals, interaction energies calculation, re-optimized all the structures in solvent have been made, which provided more valuable information to understand this radical unstrained C-C cleavage reaction. Furthermore, some important experiments involving bromoaryl substituents and medium-sized unstrained cycloketones have also been supplemented. Overall, this synthetically important paper represented an excellent work of radical-mediated unstrained C-C cleavage of cycloketones. I recommend that it can be published as it is.

Reviewer #4 (Remarks to the Author):

This manuscript has been well modified and I think it can be accepted in its current state. The predicted mechanism is reasonable to shed light on the unusual regioselectivity of C-C bond cleavage.

They have conducted extra single-point calculations with different types of functionals, including B3LYP-D3(BJ), ω B97X-D, PBE0-D3(BJ), PBE-D3(BJ), and TPSS-D3(BJ) to verified the independence of computational methods. The calculated results reveal that the trend for all the results are similar. In my opinion, there is no need to do the CCSD(T) calculations on this system. Moreover, the extra data they added have reached a satisfactory outcome, including the different theoretical levels of theory on NBO analysis and the re-optimization of all the structures in solvent.